# Solving Quantitative Reasoning Problems with Language Models

**Aitor Lewkowycz,**[*] **Anders Andreassen,**[†] **David Dohan**[†]**, Ethan Dyer**[†]**,**
**Henryk Michalewski**[†]**, Vinay Ramasesh**[†]**, Ambrose Slone,**
**Cem Anil, Imanol Schlag, Theo Gutman-Solo, Yuhuai Wu,**
**Behnam Neyshabur**[*]**, Guy Gur-Ari**[*]**, Vedant Misra**[*]

Google Research

## Abstract

Language models have achieved remarkable performance on a wide range of tasks that require natural language understanding. Nevertheless, state-of-the-art models have generally struggled with tasks that require quantitative reasoning, such as solving mathematics, science, and engineering problems at the college level. To help close this gap, we introduce Minerva, a large language model pretrained on general natural language data and further trained on technical content. The model achieves state-of-the-art performance on technical benchmarks without the use of external tools. We also evaluate our model on over two hundred undergraduate-level problems in physics, biology, chemistry, economics, and other sciences that require quantitative reasoning, and find that the model can correctly answer nearly a third of them.

## 1 Introduction

Artificial neural networks have seen remarkable success in a variety of domains including computer vision, speech recognition, audio and image generation, translation, game playing, and robotics. In particular, large language models have achieved excellent performance across a variety of natural language tasks including common-sense reasoning, question answering, and summarization [1, 2, 3, 4, 5]. However, these models have struggled with tasks that require quantitative reasoning, such as solving mathematics, science, and engineering problems [6, 7, 8].

Quantitative reasoning problems are an interesting domain of application for language models because they test the capability of models on several fronts. They require the solver to correctly parse a natural language input, potentially recall world knowledge that pertains to the problem, and apply an algorithm or series of computations to the information provided in order to arrive at a correct solution. They also require that the solver is able to correctly parse and generate precise sequences of mathematical tokens, as well as apply a computational procedure to tokens via symbolic and numerical manipulation. Finally, such problems are a proving ground for research toward robust quantitative reasoning solvers that are useful in supporting the work of humans in scientific and technical fields.

Previous research has shown that large language models achieve impressive performance on math and programming questions after training on domain specific datasets [9, 10, 11]. In this work, we apply this approach to quantitative reasoning problems, in which the model is required to provide a complete and self-contained solution, without relying on external tools. Such tasks include mathematics word problems, competition mathematics evaluations, and many problems in science and engineering.

---

[*]Equal leadership and advising contribution
[†]Equal contribution

**Question:** A line parallel to $y = 4x + 6$ passes through $(5, 10)$. What is the $y$-coordinate of the point where this line crosses the $y$-axis?

**Target:** Since the line is parallel to $y = 4x + 6$, we know that its slope is 4. Thus the equation of the line is $y = 4x + b$ where $b$ is the $y$-intercept. Plugging $(5, 10)$ into this gives $10 = 4 \cdot 5 + b \Rightarrow b = \boxed{-10}$ which is what we wanted.

- - - - - - - - - - - - - - - - - - -

**Model output:** The line is parallel to $y = 4x + 6$, which means that it has the same slope as $y = 4x + 6$. Since the line passes through $(5, 10)$, we can use the point-slope form of a line to find the equation of the line:

$$y - 10 = 4(x - 5)$$
$$y - 10 = 4x - 20$$
$$y = 4x - 10$$

Now that we have the equation of the line, we can find the $y$-coordinate of the point where the line crosses the $y$-axis by substituting $x = 0$ into the equation:

$$y = 4 \cdot 0 - 10 = \boxed{-10}.$$

**Question:** Assume that variance of first $n$ natural numbers is 10 and variance of first m even natural numbers is 16. Compute $m + n$.

- - - - - - - - - - - - - - - - - - -

**Model output:** Variance of first $n$ natural numbers is $\frac{n^2-1}{12}$. Variance of first $m$ even natural numbers is $\frac{2^2(m^2-1)}{12}$. So, $\frac{n^2-1}{12} = 10$ and $\frac{2^2(m^2-1)}{12} = 16$. Solving for $n$ and $m$, we get $n = 11$ and $m = 7$. So, $m + n = 18$.

**Question:** Prove that for every $a, b$, $b \neq a$, holds $\frac{a^2+b^2}{2} > \left(\frac{a+b}{2}\right)^2$.

- - - - - - - - - - - - - - - - - - -

**Model output:** We have

$$\frac{a^2 + b^2}{2} > \left(\frac{a+b}{2}\right)^2$$
$$\frac{a^2 + b^2}{2} > \frac{a^2 + 2ab + b^2}{4}$$
$$2a^2 + 2b^2 > a^2 + 2ab + b^2$$
$$a^2 - 2ab + b^2 > 0$$
$$(a - b)^2 > 0,$$

which is true for all $a$ and $b$.

Figure 1: Example questions with solutions from Minerva 62B. These solutions were selected by majority vote over 256 samples, or maj1@256. All questions were prompted with the same prompt (a 4-shot prompt from MATH; see appendix). **Left:** Question from the MATH dataset (Algebra 4). The model derives the solution by writing down a line equation, simplifying it, substituting $x = 0$, and solving for $y$. The model outputs valid LATEX along with natural language explanations. This detailed approach is different from the one used in the ground truth target. **Top right:** Example question from the Joint Entrance Examination Main Math 2020 exam taken each year by almost 2M Indian high-school students intended to study engineering and similar fields. **Bottom right:** Example question from the National Math Exam in Poland (May 2022). The exam is taken by approximately 270K high-school students every year.

## 1.1 Our Contribution

We present Minerva, a language model that achieves strong performance on many quantitative reasoning tasks. Our model is able to process scientific and mathematical questions formulated in natural language, and generate step-by-step solutions using correct LATEX notation. Figures 1 and 2 show example responses from Minerva to mathematics and physics questions.

Minerva is based on PaLM general language models [5] that are further trained on a high-quality dataset containing scientific and mathematical data. In particular, we start with 8B, 62B, and 540B parameter pretrained models, and continue training them on our technical content dataset. We achieve state-of-the-art performance on MATH [6], GSM8k [7], and a STEM subset of the MMLU [12] dataset, all of which include math and science questions formulated in natural language. Notably, our models achieve robust performance in a few-shot setting with no explicit training on these datasets.

The main novelty of this paper is the use of a large training dataset that juxtaposes natural language with correctly-formatted mathematical content in LATEX format. The data is collected from the arXiv preprint server and from web pages that we carefully process to minimise the loss of mathematical content. In addition to achieving state-of-the-art results, our work sets a new baseline for the performance achievable on quantitative reasoning by increasing data quality and model size.

To further increase the coverage of quantitative reasoning benchmarks, we build a dataset of over 200 undergraduate-level questions in science and mathematics from MIT's OpenCourseWare (OCW).

**Question:** A particle moves so that it is at $\left(3\sin\frac{t}{4}, 3\cos\frac{t}{4}\right)$ at time $t$. Find the speed of the particle, measured in unit of distance per unit of time.

**Target:** The parametric curve $(x, y) = \left(3\sin\frac{t}{4}, 3\cos\frac{t}{4}\right)$ describes a circle with radius 3. Furthermore, it makes a full revolution at time $t = 8\pi$. *(figure)* Therefore, the speed of the particle is $\frac{6\pi}{8\pi} = \boxed{\frac{3}{4}}$.

---

**Model output:** The speed is the magnitude of the velocity vector. Therefore, we must first find the velocity vector, which is the derivative of the position vector. We have

$$\mathbf{v} = \frac{d}{dt}\left(3\sin\frac{t}{4}, 3\cos\frac{t}{4}\right) = 3\left(\cos\frac{t}{4}, -\sin\frac{t}{4}\right)\frac{1}{4} = \left(\frac{3}{4}\cos\frac{t}{4}, -\frac{3}{4}\sin\frac{t}{4}\right).$$

Therefore, the magnitude of the velocity vector is

$$|\mathbf{v}| = \sqrt{\left(\frac{3}{4}\cos\frac{t}{4}\right)^2 + \left(-\frac{3}{4}\sin\frac{t}{4}\right)^2} = \sqrt{\frac{9}{16}\cos^2\frac{t}{4} + \frac{9}{16}\sin^2\frac{t}{4}}$$

$$= \frac{3}{4}\sqrt{\cos^2\frac{t}{4} + \sin^2\frac{t}{4}} = \frac{3}{4}.$$

Therefore, the speed of the particle is $\boxed{\frac{3}{4}}$.

Figure 2: Example question from the MATH dataset (Precalculus 4), along with the ground truth target and the Minerva solution. The model correctly takes the derivative of a position vector to compute the velocity. It then computes the norm of the resulting vector, and uses a trigonometric identity to obtain a final numerical answer. Minerva takes a completely different approach from the ground truth solution. The model output was not modified, except for a few newlines that were removed to save space.

This provides a measure of our model's quantitative reasoning abilities in a chain-of-thought context beyond a pure mathematical setting.

## 1.2 Related Works

Solving quantitative reasoning problems expressed in natural language has been an active area of study [13, 14]. Prompting language models using scratchpad [15] or chain-of-thought [16] solutions can lead them to generate step-by-step solutions to unseen problems. The GSM8k work [7] showed that training verifiers to rerank model outputs can lead to improved performance. The original version of GSM8k included special syntax for algebraic calculations, which were processed by a calculator. In this work we focus on self-contained models without access to external tools.

The standard method for evaluating language models on generative tasks is to greedily sample one solution per problem. Recent works [9, 17, 18, 19] have shown that it is advantageous to sample multiple solutions per problem, and then filter those down to a final answer. We find that majority voting [19] significantly improves performance over greedy decoding.

The work [11] includes an evaluation of davinci-002, OpenAI's latest publicly available language model, on a subset of 90 problems from the MATH dataset. Due to the focus on a subset of questions, as well as changes made to the way questions are formatted, it is difficult to directly compare our results with those of [11]. In Section 3, we compare OpenAI davinci-002 with our models under the same experimental conditions.

**Code generation.** Applying code generating models to mathematical problems has been an active area of exploration. PaLM [5] showed that a large language model with code in its training dataset can achieve good performance on a code version of GSM8k. Furthermore, the Codex model [9] can generate code solutions to MATH problems [11]. These solutions often rely on external libraries to perform mathematical operations such as solving equations or taking limits. This is a complementary approach to ours, in which we directly probe the model's ability to arrive at an answer by relying only on its own reasoning capability.

**Formal mathematics.**  Mathematics developed as a discipline based in natural language, but its axiomatic fundamentals make it possible to simulate mathematical thinking. This can be achieved using specialized programming languages that facilitate the simulation of logical and mathematical thinking using a computer, such as Coq [20], Isabelle [21], HOL4 [22], Lean [23], Metamath [24] and Mizar [25]. Work on automation of proof assistants and automated theorem provers such as E [26], leanCoP [27], and Vampire [28] has substantially benefited from integration with machine learning methods [29, 30, 31, 32, 33].

**Language models applied to formal and synthetic mathematical problems.**  Previous work trained language models to predict mathematical expressions [34, 31, 32, 35, 36, 37, 38, 39]. In turn, such a predictive model can be used to guide a proof search, as done by [32]. Large language models excel in modelling natural language, though in the case of formal languages, models that facilitate retaining information about the graph structure of a given mathematical formula, such as GNNs, are still very competitive.

**Modelling mathematics as a discipline of natural language.**  New benchmark datasets [6, 40] cover more advanced mathematical topics. For specific subtasks that pertain to understanding mathematics in natural language, such as searching for related equations, it is possible to use graph convolutional networks [41]. However for directly solving such tasks from natural language, language models are facing limited competition from other classes of models.

## 2   Training and Evaluation

### 2.1   Mathematical Training Dataset

Our models were trained on a dataset of 38.5B tokens from webpages filtered for mathematical content and from papers submitted to the arXiv preprint server. In addition, the dataset includes general natural language data, which is randomly sampled from the same dataset that was used for pretraining PaLM. Our mathematical webpage dataset was constructed by collecting pages that contain mathematical expressions in MathJax format. The pages underwent a cleaning process that removes most HTML tags but preserves natural language as well as mathematical notation, including LaTeX symbols and formatting. The result is that mathematical formulae like $e^{\pi i} + 1 = 0$ or $E = mc^2$ are presented in full to the model during training. This procedure makes it possible for the model to perform well on tasks that require calculation and symbolic manipulation. Table 1 provides a breakdown of the training dataset. See Appendix B for more details.

Table 1: Proportion of data, and number of tokens, from each source in the technical training dataset. The General Natural Language dataset is a subset of the dataset used to pretrain the model.

| Data source | Proportion of data | Tokens | Present during pretraining |
|---|---|---|---|
| Math Web Pages | 47.5% | 17.5B | No |
| arXiv | 47.5% | 21.0B | No |
| General Natural Language Data | 5% | >100B | Yes |

### 2.2   Models and Training Procedure

Our approach is to start with the PaLM pretrained decoder-only transformer language models [5] , and further train (finetune) them on our mathematical dataset using an autoregressive objective. Table 2 contains the main model and training hyperparameters. The largest model, with 540B parameters, was finetuned on 26B tokens. While this model is highly undertrained compared to the 8B and 62B models, it still achieves superior performance. Additional details can be found in Appendix C.

### 2.3   Evaluation Datasets

We mainly focus on few shot evaluation, though see Appendix E.3 for a discussion of finetuned evaluation. For evaluation, we truncate the inputs from the left to 1024 tokens and we use the model to generate up to 512 tokens. When sampling once per problem, we sample greedily. When sampling

Table 2: Model architecture and continued training hyperparameters. Model training was resumed from the pretrained PaLM models, and the number of steps quoted refers only to continued training on our technical dataset.

| Model | Layers | Heads | $d_{\text{model}}$ | Parameters | Steps | Tokens |
|---|---|---|---|---|---|---|
| Minerva 8B | 32 | 16 | 4096 | 8.63B | 624k | 164B |
| Minerva 62B | 64 | 32 | 8192 | 62.50B | 416k | 109B |
| Minerva 540B | 118 | 48 | 18 432 | 540.35B | 399k | 26B |

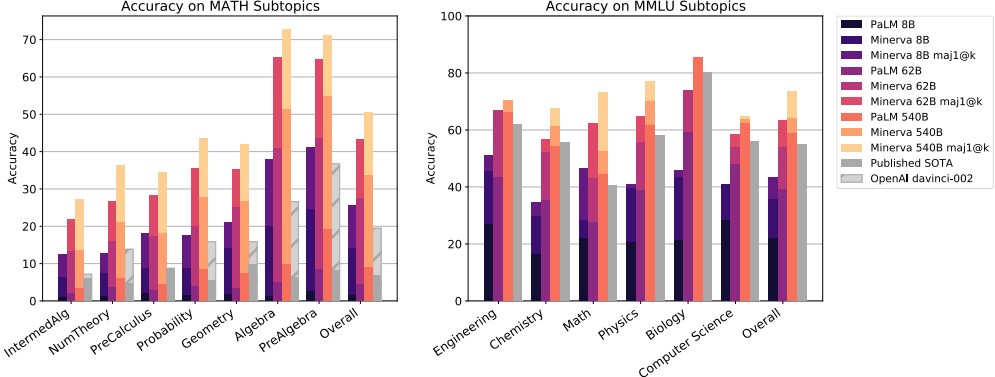

Figure 3: Performance on MATH and MMLU-STEM by subtopic. Minerva achieves state-of-the-art results on both datasets. maj1@k denotes evaluations where $k$ samples were generated for each problem and only the majority vote (most common answer) was selected [19]. For MATH, $k = 256$ for Minerva 8B and 62B, and $k = 64$ for 540B. For MMLU-STEM, $k = 16$. davinci-002 is the latest publicly available model from OpenAI.

multiple times per problem we use nucleus sampling [42] with temperature $T = 0.6$, $p = 0.95$. For generative tasks, the model produces a chain-of-thought answer and demarcates a final answer. We evaluate a solution as correct if the final answer matches the ground truth solution, independent of the quality of the chain-of-thought preceding it. To evaluate correctness, we parse the final answers and compare them using the SymPy library [43]. This is done in order to correctly identify answers that are mathematically equivalent such as $1/\sqrt{3}$ and $\sqrt{3}/3$. See Appendix D.1 for further details.

The existing datasets on which we focus are:

- MATH: a dataset of 12K middle school and high school mathematics problems [6]. Problem statements are written in LaTeX. We prompt the model with a fixed 4-shot prompt (listed in Appendix D.2). This prompt includes four random examples from the training dataset whose ground truth targets are not too long.

- GSM8k: middle school math word problems [7]. Models are evaluated using the chain-of-thought prompt from Wei et al. [16]. Previous models evaluated on GSM8k made use of an external calculator. In this work, our model does not have access to any external tools.

- MMLU-STEM: subset of the MMLU dataset [12] focused on science, technology, engineering, and mathematics (STEM). For the original version, we use the 5-shot prompt from the development set for each task. We also consider chain-of-thought prompting for this task, where we prompt the model with examples that include step-by-step solutions. We use a multiple-choice version of the MATH prompt for topics that involve mathematical reasoning, and add step-by-step solutions to the standard 5-shot prompts for the rest of the topics. See Appendix G for more details.

## 2.4 Undergraduate-Level STEM Problems

To evaluate the scientific reasoning capabilities of Minerva, we harvested a set of STEM problems at the undergraduate level, most of which involve multi-step reasoning, which we refer to in this paper

Table 3: **Model performance on several quantitative reasoning datasets.** For majority voting we use $k = 256$ (64 for 540B) samples for MATH, $k = 64$ for OCWCourses, $k = 100$ (40 for 540B) for GSM8k and $k = 16$ for MMLU-STEM. The PaLM GSM8k results do not use a calculator and were reported in [5]. We evaluated datasets that did not have published results on recent models on OpenAI davinci-002. Despite MMLU-STEM being a multiple choice task, we can apply majority vote by prompting the model to generate a rationale prior to the final answer, sampling multiple times, and then using majority vote on the final answers. Superscripts denote results that are quoted from previous work: [a] GPT-2 [6], [b] PaLM 540B `maj1@40` [19], and [c] Chinchilla [44].

| | MATH | OCWCourses | GSM8k | MMLU-STEM |
|---|---|---|---|---|
| PaLM 8B | 1.5% | 1.5% | 4.1% | 22.0% |
| Minerva 8B | 14.1% | 7.7% | 16.2% | 35.6% |
| Minerva 8B, `maj1@k` | 25.4% | 12.5% | 28.4% | 43.4% |
| PaLM 62B | 4.4% | 5.9% | 33.0% | 39.1% |
| Minerva 62B | 27.6% | 12.9% | 52.4% | 53.9% |
| Minerva 62B, `maj1@k` | 43.4% | 23.5% | 68.5% | 63.5% |
| PaLM 540B | 8.8% | 7.1% | 56.5% | 58.7% |
| Minerva 540B | 33.6% | 17.6% | 58.8% | 63.9% |
| Minerva 540B, `maj1@k` | **50.3%** | **30.8%** | **78.5%** | **75.0%** |
| OpenAI davinci-002 | 19.1% | 14.8% | - | - |
| Published SOTA | 6.9%[a] | - | 74.4%[b] | 54.9%[c] |

as OCWCourses. Using publicly-available course materials offered by MIT (OpenCourseWare), we collected problems with automatically-verifiable solutions (either numeric or symbolically verifiable via `SymPy`) from courses including "solid-state chemistry", "information and entropy", "differential equations", and "special relativity." These problems were processed by contractors to be self-contained and to have a clearly-delineated final answer. Problems asking for a proof or open-ended short answer were not included. In total we curated 272 problems, 191 of which have numeric solutions and 81 have symbolic solutions. In Appendix F, we detail the contributions from each course, and the process of converting these course materials into a format suitable for processing by language models. We also provide the text of all problems. We plan to release these as part of an open-source dataset which will be detailed in an upcoming manuscript.

## 2.5 Inference-Time Techniques

We find that we can considerably outperform greedy decoding by sampling $k > 1$ solutions (with a non-zero temperature) and selecting one using majority voting [19]. This consists of grouping predictions with respect to their final answer and selecting the most common answer. We denote this as `maj1@k`, following [17]. A variation of this algorithm, denoted `majn@k`, involves selecting the $n$ most common answers. Intuitively, the reason majority voting improves performance is that while there are many ways to answer a question incorrectly, there are typically very few ways to answer correctly. In other words, the truth is a Schelling point.

Contrast majority voting with `pass@k`, where a task is considered solved if any single sample solves it out of $k$ samples. See Section 4.2 for more details on `pass@k` performance. In Appendix E.1, we report on how performance depends on $k$ for different metrics. We find that while `pass@k` continues to improve as $k$ is increased, majority voting performance saturates faster: 97% of the large $k$ accuracy is achieved at $k = 64$ for MATH and $k = 16$ for GSM8k. This is likely because majority voting selects the most common answer in the modeled distribution, and the error of this estimate decreases with increasing $k$. This is in contrast to `pass@k` where the performance improvement comes from the tail of the distribution, which can keep improving as $k$ is increased.

Log-likelihood is another metric that can be used to rerank samples. We found that majority voting performs significantly better than log-likelihood reranking (see Appendix E.2).

# 3 Results

Table 3 summarizes the results for Minerva models and other models, on the evaluation datasets described in Section 2.3. Figure 3 presents a breakdown of the MATH dataset results by subtopic. For MMLU evaluations, unless otherwise noted, performance is measured by using the standard 5-shot prompt per topic and picking the answer with the highest score. When evaluating MMLU with majority voting, we sample $k = 16$ model answers using a chain-of-thought prompt.

We present model output samples in Figures 1 and 2, and additional output samples are listed in the Appendix. In addition, we evaluated Minerva 62B on the National Math Exam in Poland and found that it achieves a score of $57\%$, which happened to be the national average in 2021 [45, p. 23]. The 540B model achieves $65\%$.

We include results on the latest publicly available language model from OpenAI, davinci-002, evaluated using the OpenAI API with temperature set to the official recommendation ($T = 0.2$). The combination of training data, scale and inference techniques yields state of the art results on all the technical tasks that we considered. For all tasks (with the exception of GSM8k), the improvement with respect to previous results is considerable.

While our main focus is on few shot evaluation, we also tried to finetune Minerva on MATH; this did not yield any improvement; interestingly, however, finetuning PaLM on MATH did give a significant improvement. Further details can be found in Appendix E.3.

## 3.1 Basic arithmetic

In Appendix I, we study the performance of Minerva 540B on simple arithmetic tasks. The model achieves over $80\%$ accuracy on 10-digit addition and over $20\%$ accuracy on 18-digit addition.

# 4 Performance Analysis

## 4.1 Model Mistakes

To better understand the types of mistakes our models make, we compare the performance of Minerva 8B and Minerva 62B on 216 problems with high confidence majority decisions of both models. Specifically, we selected examples where the top answer received at least $15\%$ of votes, and that either Minerva 8B was correct and Minerva 62B was incorrect (15 samples), or vice versa (201 samples). The categories and examples for each category are described in Appendix J.2.

As shown in Table 4, the prevailing errors of the 8B model were related to incorrect reasoning or calculations. Many of the calculation errors were relatively benign arithmetic mistakes. Solutions that were too short were relatively rare (in these cases, the model immediately produces an incorrect answer without any intermediate reasoning steps). Finally, in a few cases, the model hallucinates an equation or mathematical fact that is not real.

In the samples where the 62B model was incorrect, the dominating failure modes were again incorrect reasoning and incorrect calculations. In summary, we find that the 62B Minerva model retains most of the skills of the 8B model and improves upon both reasoning and calculation robustness.

Table 4: Failure modes of the 8B Minerva model, out of 201 samples which the 62B model solved correctly and the 8B model did not.

| Type of mistakes | Occurrences | Type of mistakes | Occurrences |
|---|---|---|---|
| Incorrect reasoning | 82 | Incorrect calculation | 70 |
| Misunderstands question | 22 | Uses incorrect fact | 16 |
| Solution too short | 4 | Hallucinated math objects | 4 |

## 4.2 False Positives

In our approach to solving quantitative reasoning problems, we are able to automatically verify whether the final answer to a problem is correct, but we do not have an automatic way to verify the

model's chain of reasoning. This leaves open the possibility of false positives: samples which have the correct final answer, but for which the reasoning is incomplete or incorrect.

We selected 100 random questions from MATH (20 per difficulty level), along with answers sampled at zero temperature from the 62B model. We then manually inspected the answers to determine the false positive rate, which is the ratio between number of false positive examples and number of examples for which the final answer is correct; for difficulty levels 1, 2, 3, 4, and 5, the false positive rates are $< 5\%$, $10\%$, $< 5\%$, $15\%$, and $30\%$, respectively. Averaging these per-level false positive rates (weighted by true positive rates), the overall false positive rate is estimated to be $8\%$. Although this overall rate is low, we find that it does increase with difficulty level.

Our focus on `pass@1` and majority voting as the primary evaluation metrics is due in part to the fact that they are less susceptible to false positives than `pass@k` [17]. While the `pass@256` accuracy is $84.5\%$ for the 62B model, false positives account for part of it. We inspected the samples that failed in majority voting but passed on `pass@k` due to a single correct answer, and estimate the false positive rate for `pass@256` to be 30% among samples selected in this way. After removing false positives, we estimate that the `pass@256` accuracy at greater than 68%; see Appendix J.3 for details.

## 5 Memorization

A central question in interpreting Minerva's solutions is whether performance reflects genuine analytic capability or instead rote memorization. This is especially relevant as there has been much prior work indicating that language models often memorize some fraction of their training data [46, 47, 48]. When examining model solutions, we find that memorization of intermediate facts, such as numerical values of square roots or trigonometric identities, are crucial elements of model solutions. We would like to investigate a strong form of memorization, where model performance is a result of memorizing the explicit problems and solutions in our evaluation set, but also a weaker form, where the model has memorized alternate answers to the same questions.

In order to evaluate the degree to which our models solve problems by recalling information memorized from training data, we conduct three analyses on the MATH dataset. First we directly search for problems and solutions in our training corpus. Next, we generate modified versions of problems and evaluate our models' robustness to these changes. Finally, we measure the degree of overlap between the ground truth solutions and solutions generated by our model and measure the effect of this similarity on model performance. Overall, we find little evidence that the model's performance can be attributed to memorization.

### 5.1 Training and Evaluation Dataset Overlap

We selected the problems for which our 62B parameter model produced a correct answer, and filtered them to the 100 problems with the highest majority vote score, expecting that problems with a high majority vote score are more likely to have been memorized. For each of these question-answer pairs, we compute the BLEU score across chunks of 500 characters in our Math Web Pages dataset (a histogram of the BLEU scores is shown in Appendix Figure 9). We then manually inspect the 250 documents with the highest BLEU scores. While many of the top matches were from homework help sites with math questions and solutions, none of the questions matched the questions in the subset of MATH under consideration. We have included these 250 segments in Appendix K.1. We note that some problems from MATH can be found on the web. Nevertheless, this analysis concludes that these problems did not make it through our data collection process.

### 5.2 Performance on Modified MATH Problems

To further investigate memorization, we randomly selected twenty problems which the 62B model answered correctly under majority voting. We manually modified each problem either by introducing minor changes to problem wording (framing) or by changing the numbers which appeared in the problem and modifying the solution accordingly. We then compared the accuracy over sampled solutions before and after the modification. Results are shown in Figure 4. In both cases the accuracy before and after modifications are correlated, with no clear bias in favor of the original formulation. This is suggestive of minimal memorization. The modified problems are listed in Appendix K.2.

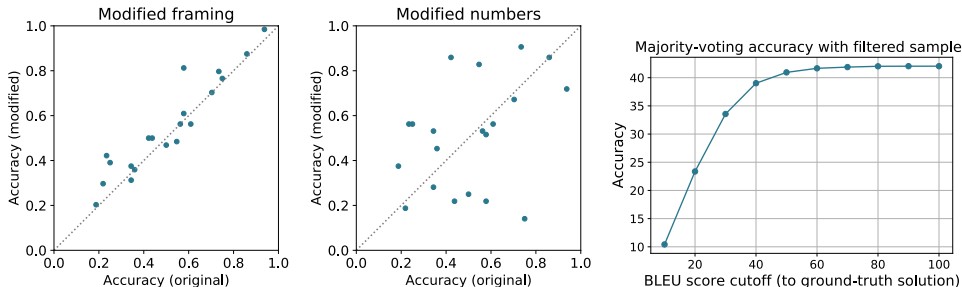

Figure 4: **Results indicating lack of memorization on MATH. Left, Center:** Accuracy of original questions from the MATH dataset and their modified versions. Each point represents a question. The $x$ axis is accuracy on the original question, and the $y$ axis is accuracy on the modified one. **Right:** Majority vote accuracy, computed only on samples with BLEU score to the ground truth solution less than or equal to the $x$-axis value.

### 5.3 BLEU Score Between Ground Truth and Generated Solutions

We seek to detect memorization of solutions by computing BLEU score between ground truth answers and model generated answers. We use the 62B model and analyze 256 samples per problem in the MATH dataset. First, we compute overlap statistics for all correct samples. We find that 160 out of 5,000 test questions have a sample with a BLEU score greater than or equal to 80 (see Appendix K.3). We note that they tend to be short solutions. To understand the effect of answer similarity on performance, we remove model samples above a certain BLEU score threshold, and recompute the majority vote accuracy. We find that majority vote performance is robust even down to relatively low similarities (see Figure 4), indicating that performance cannot be attributed to model outputs that are very similar to ground truth answers.

## 6 Conclusions and Discussion

In this work, we take an approach to quantitative reasoning that relies on solving problems using mathematical reasoning expressed in natural language. We show that by training a large language model on a high quality mathematical dataset, we are able to achieve strong performance on tasks that require logical reasoning, numerical calculation, and symbolic manipulation. Our model does not make use of external tools, and at inference time relies exclusively on autoregressive sampling to achieve this performance. Complementary approaches to quantitative reasoning include code-generating models and formal methods. These are all different routes toward a common goal: an agent that can reason about and solve quantitative problems.

### 6.1 Limitations of Our Approach

Our approach to quantitative reasoning has several limitations. First, we have no automatic way of verifying the correctness of the model's answers. This is in contrast to formal approaches, for which automatic verification is intrinsic. Second, our model has no access to external tools such as a calculator or a Python interpreter. It is therefore limited in its ability to perform quantitative reasoning tasks that require complicated numerical calculations. Third, because our model was trained on a large amount of data, we have little direct control over the specific capabilities that the model acquired.

Our inference time strategy also has limitations. Majority voting requires generating many samples and selecting the most common final answer. This can be computationally costly, and importantly only works for problems with well defined final answers, and so is less well suited for domains such as theorem proving or open-ended scientific reasoning.

### 6.2 Societal Impact

Artificial neural networks capable of solving quantitative reasoning problems in a general setting have the potential of substantial societal impact. Minerva, while a step in this direction, is still far from achieving this goal, and its potential societal impact is therefore limited. The model's performance is still well below human performance, and furthermore, we do not have an automatic way of verifying

the correctness of its outputs. If these issues could be solved, we expect the impacts of this model to be broadly positive. A direct application could be an accessible and affordable math tutor which could help improve educational inequalities.

## 7 Acknowledgments

We thank David Andre, Jacob Austin, Maarten Bosma, Aakanksha Chowdhery, Sergey Ioffe, Colin Raffel, Charles Sutton, Christian Szegedy, Stanislas Polu, and Harrison Edwards for helpful discussions.

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
