# OpenReview forum: "Solving Quantitative Reasoning Problems with Language Models"
_NeurIPS.cc/2022/Conference — NeurIPS 2022 Accept_

### Official Review · Reviewer_h3bW · 2022-07-06

**Rating:** 6
**Confidence:** 5
**Soundness:** 3 good
**Presentation:** 3 good
**Contribution:** 2 fair

**Summary:**

The paper focuses on the problem of quantitative reasoning in large language models, and introduces Minerva, a large language model that is pretrained on general language data and then further trained on a dataset containing scientific and mathematical data.

The training data is collected from arXiv and web pages that were preprocessed to conserve mathematical content. The authors then use this data to finetune pretrained decoder-only transformer LMs (8B and 62B parameter models) using an autoregressive objective. During inference time, the paper suggests sampling k > 1 solutions and selecting one using majority voting. The authors show empirically that this approach considerably outperforms greedy decoding.

The authors compare their models against previous works on the MATH, GSM8k, and MMLU-STEM datasets, and also build a dataset (OCWCourses) of 272 undergraduate questions in science and mathematics to evaluate the model's quantitative reasoning abilities in a chain-of-thought context. The authors report that the Minerva 62B significantly outperforms state-of-the-art results on the MATH dataset, and achieves similar performance on MMLU-STEM. Furthermore, the authors conduct analyses to understand the types of mistakes the models make, the effect of false positives (cases where the answer is correct but the chain of reasoning is not), and whether the model is memorizing the answers. They find that most of the errors are incorrect reasoning and incorrect calculations, that false positive rate increases with difficulty level (with an estimated average of 8% on the MATH dataset), and that there is little evidence that the model's performance can be attributed to memorization.





**Questions:**

1- Are the "BaseModel" models used publicly available? Appendix B does not contain sufficient information for reproducibility.
2- Table 4 only reports the distribution of failure modes of the 8B model, and the text only mentions the dominating failure modes for the 62B model. What is the distribution for the 62B model, where the 8B model solved correctly and the 62B model did not? Is there a further breakdown of incorrect reasoning?

**Limitations:**

The authors discuss the limitations and potential societal impact of their work.

**Strengths And Weaknesses:**

**Strengths**

The topic of quantitative reasoning with large language models is of interest to the research community. By finetuning pretrained LMs on large math and science datasets and comparing against models that were not trained with such data, the paper highlights the importance of data quality for achieving quantitative reasoning abilities. Furthermore, the authors build a new dataset for evaluating quantitative reasoning, and conduct thorough evaluation of the type of mistakes the models make, highlighting the issue with false positives, and investigating the impact of memorization on performance.

**Weaknesses**

Although the authors mention that the dataset is the main novelty of their paper (line 45), there isn't sufficient discussion of it in the paper and appendix. Will this approach scale to more complex math and science problems? Are further gains expected with larger models? Furthermore, what is the breakdown of papers by area/topic? How are diagrams (as mentioned in the intro) processed? What are the examples of heuristics used to process the pages as mentioned in the appendix? etc.

---

> ### Author Response · Authors · 2022-08-02
> **Response to reviewer h3bW**
>
> We thank the reviewer for their review and feedback.
> Regarding the overall contribution, we refer the reviewer to the general comment to all reviewers, and also to the response to reviewer iAwH highlighting the importance and implications of our work.
>
> We also address the reviewer’s specific questions below:
>
> > “Are the "BaseModel" models used publicly available?”:
>
>
> The “BaseModel” is not publicly available, and as mentioned in footnote 1 on page 2, details about the BaseModel are omitted for the purpose of anonymization. Once anonymization is no longer an issue, the name of the BaseModel will be revealed and we will add the appropriate citations describing the BaseModel.
>
> > “Table 4 [...] What is the distribution for the 62B model, where the 8B model solved correctly and the 62B model did not? Is there a further breakdown of incorrect reasoning?”
>
>
> We manually inspected a set of problems that the 8B model solved correctly and the 62B model did not, and we observed that similar to the reverse setup, the main failure mode is incorrect calculation and reasoning. Due to the similarity, we did not do a systematic study on a sufficient number of examples to report statistics in this case.
>
> > “[ …] dataset is the main novelty of their paper (line 45), there isn't sufficient discussion of it in the paper and appendix.”:
>
>
> We are unfortunately unable to release the full dataset and exact scripts that generated the dataset, but we have provided some details with the general idea of the sources and processing that was used. Since our submission actually emphasizes the capabilities of the models trained on our dataset, and not the dataset itself, we have reworded this sentence in the paper to say: “The main novelty of this paper is the use of a large training dataset that juxtaposes natural language with correctly-formatted mathematical content in LaTeX format.”.
>
> > “Will this approach scale to more complex math and science problems? Are further gains expected with larger models?”
>
>
> MATH contains 5 difficulty levels, and we do find that more difficult problems are less likely to be solved by Minerva, as one would expect, and we do expect larger models to show better performance. In the updated manuscript, we have added our results from a 540B parameter model, which achieves SOTA performance on all the benchmarks studied in this submission.
>
> > “How are diagrams (as mentioned in the intro) processed?”:
>
>
> We use all diagrams which are scripted using tikz, asymptote, and other rendering frameworks, directly as written, while we remove diagrams which are in image format, such as jpeg, pdf or similar. We have updated the wording in the intro to make this explicit. Thank you for pointing this out!

---

### Official Review · Reviewer_9sZQ · 2022-07-09

**Rating:** 2
**Confidence:** 3
**Soundness:** 3 good
**Presentation:** 2 fair
**Contribution:** 2 fair

**Summary:**

The authors fine-tune large PLMs (pre-trained on a corpus of scientific and mathematical text) on a new quantitative reasoning dataset, evaluating on existing benchmarks and a new, custom benchmark. The main contribution is the formulation of these datasets, which allows for sota on MATH and strong performance on other benchmark. Notably, this approach relies exclusively on a language model, which has no outside access to libraries, calculators, or other tools in performing the task.

**Questions:**

- The claim in L175-177 seems unfounded to me. Why wouldn't you be sure you haven't found a local minimum with the fine-tuned model by training on the wrong dataset, whereas you haven't with the base model?
- In L57-58, I cannot parse "Prompting language models... unseen problems". Do you mean to say "apply" instead of "output"?
- I cannot tell whether the pre-trained model, on which Minerva is based, is a contribution of this paper or another paper?
- I don't understand the need for the OCW benchmark introduced, in addition to MATH, GSM8k, and others. Which part of the new dataset do these more established benchmarks lack?


**Limitations:**

The authors rightly identify 3 limitations of their method. There is nothing set in stone, however, about these limitations (one could engineer a dataset to give specific capabilities to a model, or use a model that had access to outside tools, etc), and the authors do not discuss why they chose the model, data, etc. that they did __with respect to__ these limitations.

**Strengths And Weaknesses:**

Strengths
- The most obvious strength of the paper is the considerable improvement over SOTA scores, although I'm quite surprised at how low such scores are, especially when GPT-3 also so exceeds them.
- I think the majority voting idea is really cool, since sampling obviously provides a limited view of the branching probability tree characterizing the language model's processing of the original prompt, and majority rule arguably offers a more fulsome picture of the outputs. The obvious problem with it is that cost scales linearly.
- Checking the answers using sympy is a clever way to bin different expressions of answers.

Weaknesses
- The main problem with the paper, in my view, is that the training data is the main contribution of the paper, but I cannot tell what it is like. Basically the only thing I can tell is that it contains mathematical expressions. Fig. 1 and Fig. 2 show the evaluation data, which is already outlined in other work, and I wonder why that space wasn't used to explain this paper's training corpus. At one point, the authors mention "equations and diagrams" juxtaposed with natural language, but how would a language model process diagrams, or how would they be converted to text? I have many more questions, and they might be answered in the appendix, which wasn't available to me, but this lack of clarity seems to me to preclude the publication of this paper at NeurIPS.
- The benchmark introduced may be too small to be meaningful, and I don't think the paper makes it clear how representative it is. Furthermore, I worry, when it includes questions from "‘solid-state chemistry’, ‘information and entropy’, ‘differential equations’, and ‘special relativity'", that quantitative reasoning is not what we are testing, but specific scientific knowledge that makes use of quantitative reasoning.
- Doesn't reference EQUATE (Ravichander et al. 2019), one such recent, decently-cited quantitative reasoning benchmark.
- The authors say in L51 that "existing benchmarks are limited with respect to quantitative reasoning", and I'd like to know what they mean by that. I couldn't find any corresponding discussion.

---

> ### Author Response · Authors · 2022-08-02
> **Response to reviewer 9sZQ**
>
> Thank you for your valuable feedback; we are happy that you find the majority voting interesting and appreciate the strength of the model in answering questions from MATH. We have addressed all your concerns by adding new results, updating the paper and providing answers to your questions. In light of that, we hope you consider recommending our paper for acceptance.
>
> First, let us address the concerns about our training dataset, specifically:
>
> > The main problem with the paper, in my view, is that the training data is the main contribution of the paper, but I cannot tell what it is like.
>
> In the appendix, section A, we describe in detail how our dataset is created and what documents it comprises.  Specifically, section A.1 explains the selection and processing of the arXiv paper dataset, and section A.2 describes the selection and processing of the mathematical web pages dataset.
>
> > At one point, the authors mention "equations and diagrams" juxtaposed with natural language, but how would a language model process diagrams, or how would they be converted to text?
>
> Thank you for pointing this out; this was poorly worded, as there are no diagrams in the dataset (unless they are already specified by text, for example as tikz or asymptote).  We have rewritten this sentence as follows:
>
> “The main novelty of this paper is the use of a large training dataset that juxtaposes natural language with correctly-formatted (in LaTeX) mathematical content.”
>
> If you have other questions about the dataset, which aren’t answered by the Appendix sections, we would be happy to address them.
>
> **Regarding the comment about the OCWCourses dataset**
>
> > The benchmark introduced may be too small to be meaningful, and I don't think the paper makes it clear how representative it is.
>
> All of the problems in this dataset are available as a .tex file in the supplementary material; please see the file ocw_courses_problems.tex.  Regarding the concern about size, we are in the process of expanding this dataset and will release a larger dataset comprising these problems once completed.
>
> > Furthermore, I worry, when it includes questions from "‘solid-state chemistry’, ‘information and entropy’, ‘differential equations’, and ‘special relativity'", that quantitative reasoning is not what we are testing, but specific scientific knowledge that makes use of quantitative reasoning.
>
> This is true; as these questions were chosen to be representative of undergraduate STEM problems, some of them require quantitative reasoning and others rely more on knowledge.
>
> **Regarding EQUATE**
>
> Thank you for bringing this dataset to our attention.  We have evaluated our model on this dataset as well; the details are in the Appendix, section G.
>
> **Responses to minor questions**
>
> > The claim in L175-177 seems unfounded to me. Why wouldn't you be sure you haven't found a local minimum with the fine-tuned model by training on the wrong dataset, whereas you haven't with the base model?
>
> We have removed this claim, and just kept the description of the empirical result.
> > In L57-58, I cannot parse "Prompting language models... unseen problems". Do you mean to say "apply" instead of "output"?
>
> We have reworded this sentence to be clearer:
> Prompting language models using scratchpad (https://arxiv.org/abs/2112.00114)  or chain-of-thought ()​https://arxiv.org/abs/2201.11903) solutions can lead them to generate step-by-step solutions to unseen problems.
> > I cannot tell whether the pre-trained model, on which Minerva is based, is a contribution of this paper or another paper?
>
> This pre-trained model is a contribution of another paper; we will cite this in the deanonymized, camera-ready version of the paper.
>
> > I don't understand the need for the OCW benchmark introduced, in addition to MATH, GSM8k, and others. Which part of the new dataset do these more established benchmarks lack?
>
> OCW tests quantitative reasoning and scientific knowledge in a free-response setting on a more challenging class of problems, designed for undergraduate students rather than middle school and high school students. By contrast, MATH and GSM8k focus specifically on mathematics up to high-school difficulty, and MMLU is a multiple-choice benchmark.
>
> We hope that this response has addressed your concerns about recommending to accept this work.

---

### Official Review · Reviewer_iAwH · 2022-07-11

**Rating:** 6
**Confidence:** 4
**Soundness:** 3 good
**Presentation:** 4 excellent
**Contribution:** 3 good

**Summary:**

State-of-the-art language models have generally struggled with tasks that require quantitative reasoning, such as solving mathematics, science, and engineering problems at the college level. This paper introduced Minerva, a large language model pretrained on general natural language data and further trained on technical content. The authors conducted experiments in a few-shot setting across a range of tasks that require quantitative/numerical/symbolic reasoning, including one they manually collected for better evaluation (OCWCourses). The results demonstrated that it achieved strong performance on all the datasets and set state-of-the-art
 performance on the MATH dataset. Through an error analysis, they also found the two most common cases in which their model made wrong predictions are *Incorrect reasoning* and *Incorrect calculation*.




**Questions:**

n/a

**Limitations:**

The authors have sufficiently addressed the limitations in the paper.

**Strengths And Weaknesses:**

- Originality: The idea of further training a pretrained language model on a math-heavy corpus is quite straightforward yet effective (as can be seen from the results).

- Quality: The proposed approach is technically sound and the experimental results showed that it outperformed the previous language model on MATH and OCWCourses by a large margin. However, I don't think there are many insights I can draw from this paper. My main takeaway about this paper is that we can achieve much better performance on quantitative/numerical/symbolic reasoning by further training a pretrained language model on high-quality corpus containing equations and diagrams. What if we just train a language model from scratch using the same corpus? What is the role of the general language data played here? I think answering such questions could make this paper stronger.

- Clarity: This paper is well-organized and easy to follow.

- Significance: This paper achieved strong results across a range of different datasets and the reasoning skill demonstrated by the model is amazing. As pointed out by the authors that one of the main challenges is to automatically verify the correctness of the generated reasoning chains/answers, I believe this work would prosper the work in such directions.

---

> ### Author Response · Authors · 2022-08-02
> **Response to reviewer iAwH**
>
> Thank you for useful feedback! We have updated the paper (please also see our general comment) and provided responses to your review. Please let us know if this sufficiently addresses your concerns.
>
> Regarding your comments on originality and quality, we want to highlight that our work has very important implications regarding two main areas of disagreement among researchers working on reasoning:
>
>  - **The role of scaling**: Reasoning problems were perhaps the most important group of problems that seemed not to benefit significantly from scaling. For example, Hendrycks et al. 2021 evaluated GPT-3 on the MATH dataset and estimated that “Assuming a log-linear scaling trend, models would need around 10^35 parameters to achieve 40% accuracy on MATH, which is impractical.” Given these results, many experts were skeptical of the ability of language models to achieve accuracy 50% any time soon, especially without making fundamental architectural changes that facilitate symbolic manipulation. Our largest model reported in our revision reaches the 50% accuracy milestone (please see our general comment highlighting new results and updates to the paper).
>
>  - **Formal vs. informal**: Formal mathematics has historically been the main approach to using AI for mathematical proving and problem solving. Our work provides strong evidence that even though the informal direction is not able to benefit from tools such as automatic verification, it is a very promising direction given our access to a large corpus of technical data and the abilities of large language models to perform multi-step reasoning using scratchpad/chain-of-thought.
>
> We also want to emphasize that while the story appears to be simple, successful training of large models involves a significant amount of work and there are many decisions that could make or break them. For example, careful processing of technical data was an important element (please see appendix C on “Training Dataset Details”).
>
> Regarding “I don't think there are many insights I can draw from this paper”, we believe while there are many analyses and useful insights in the paper (and we have added more in the revision), the conference page limit and the significance of the main result might make it difficult to draw attention to them. Based on your feedback, we have improved the presentation of the paper. Here, we highlight some of the analyses and useful insights in the paper :
>
>  - Significant role of scratchpad/chain-of-thought + majority voting in performance (Figure 3 and Table 3)
>  - We study the dependence of performance on the number of generated samples per question (Table 6 and Figure 5).
>  - We compare majority voting to log-Likelihood reranking and show that majority voting has superior performance (Table 7).
>  - We compare Few Shot, Custom prompt finetuning, Random prompt finetuning and No prompt finetuning and show that while finetuning our base model outperforms its shot prompting, Minerva’s few shot performance is superior to finetuned performance. We hypothesize that this can be attributed to the quality of our technical data used for training Minerva (Table 8).
>  - We analyze the asymptotic behavior of majority voting at large k (Section D.4) .
>  - Investigating the mistake types and classifying them into several categories and showing that 75% of them are incorrect reasoning and incorrect calculations (Table 4)
>  - We show that while the false positive rate is low, it increases with the difficulty level
>  - In a memorization sanity check, we modify MATH questions and show that Minerva’s accuracy before and after modifications are correlated, with no clear bias in favor of the original formulation (Figure 4).
>  - Minerva has great performance on arithmetic tasks but the performance still degrades with number of digits (Figure 7)
>  - We show that SymPy equivalence would only marginally improve the performance (Table 5)

---

### Official Review · Reviewer_8LQc · 2022-07-11

**Rating:** 7
**Confidence:** 4
**Soundness:** 4 excellent
**Presentation:** 4 excellent
**Contribution:** 3 good

**Summary:**

This paper introduces a large language model and new training and evaluation dataset for quantitative reasoning problems. It first overviews existing methods and resources for quantitative reasoning, then discusses the datasets collected for training and evaluation, and finally provides comprehensive results and analysis of a model that is finetuned on the collected training data, finding that in comparison to existing models, the proposed large language model performs well.

**Questions:**

* Clarification question: Table 1 lists tokens in the entire dataset, right? Not the amount sampled for the technical training dataset? Otherwise the proportions don't make sense.
* Is the general NL data for the technical dataset sampled randomly from the full general NL dataset?
* Why truncate inputs from the left? Besides fitting in the limited transformer input. How often does this happen?
* What does it look like for reasoning to be wrong but the answer to be right? It's a bit surprising that the false positive rate is 30% in difficulty 5 examples -- without knowing what those examples look like, I find it very surprising that there are so many false positives.
* How many times each are the 20 examples in 5.2 modified to test memorization?

**Limitations:**

I'm wondering about copyright/consent issues when it comes to scraping/re-releasing data from math webpages and ArXiv. Are you planning to release the training data?

Similarly, details on the contract work used to clean the dataset -- typically crowdsourcing contributions will have (at least in the appendix) details on compensation, etc. but I couldn't find that.

**Strengths And Weaknesses:**

Strengths:

* Good overview on challenges associated with quantitative reasoning
* Very comprehensive error analysis (and examples in appendix)
* Great analysis of the model robustness wrt. memorization

---

Weaknesses:

* The main contribution here is collecting (scraping and filtering) a dataset, and fine-tuning a large model on it. By itself, I am not sure this is a huge contribution. However, the analysis in the latter half of the paper is quite comprehensive.
* I was confused about the majn@k format in Fig 1 and 2.3 until section 2.5. Would help to clarify in the Fig 1 caption and 2.3 that it's majority voting on the final answer, not the whole generated text (how does this work for something like a proof, though, that's inherently a chain of reasoning?).
* I'd like to see more discussion on the limitations of existing quantitative benchmarks. I was expecting this in the related work section but couldn't find anything (just briefly mentioned on L86).

---

> ### Author Response · Authors · 2022-08-02
> **Response to reviewer 8LQc**
>
> Thank you for taking the time to read our work, for the positive comments and for the suggested clarifications and improvements!
>
> **Weaknesses**
> > The main contribution here is collecting (scraping and filtering) a dataset, and fine-tuning a large model on it. By itself, I am not sure this is a huge contribution. However, the analysis in the latter half of the paper is quite comprehensive.
>
> We are glad that you appreciate the analysis! We agree that this paper largely relies on existing techniques. We want to emphasize that one of the major takeaways from this work is precisely that, contrary to some expectations in the field, major performance improvements on quantitative reasoning tasks can be achieved solely with currently available techniques.
>
> > I was confused about the majn@k … would help to clarify in the Fig 1 caption and 2.3
>
> Thanks for the suggestion. We have now clarified this in the captions for Figure 1 and Figure 4.
>
> > how does this work for something like a proof, though, that's inherently a chain of reasoning?
>
> This is an insightful question, and indeed a limitation of majority voting is that it can only be applied in contexts with a clear final answer. Other sample re-ranking techniques, such as verifiers [Cobbe et al. (2021)], can perhaps be repurposed during the generation process to help in more open ended problem solving, such as theorem proving. This is an active area of future research. We have now emphasized this limitation of majority voting in section 6.1 Limitations.
>
> > I'd like to see more discussion on the limitations of existing quantitative benchmarks.
>
> We have modified the discussion at the end of the Our Contributions section. The quantitative benchmarks we are aware of that present chain-of-thought rationale, rather than just final answers, focus on mathematics. The OCW dataset helps broaden this to cover more STEM fields.
>
> **Questions**
> > Table 1 lists tokens in the entire dataset, right? Not the amount sampled for the technical training dataset? Is the general NL data for the technical dataset sampled randomly from the full general NL dataset?
>
> Yes the token count is the entire dataset. The general NL data is sampled randomly from the larger dataset. We have clarified this now in the text.
>
> > Why truncate inputs from the left? Besides fitting in the limited transformer input. How often does this happen?
>
> The only reason to truncate is what you describe, so that prompt, question, and answer can all fit in the transformer window. This happens in 3% of MATH examples.
>
> > What does it look like for reasoning to be wrong but the answer to be right? It's a bit surprising that the false positive rate is 30% in difficulty 5 examples.
>
> To make it easier to inspect, we have now included all of the false positive examples analyzed in the supplementary file minerva_false_positives.json. Anecdotally, there are a few different types of false positives. Some false positives occur when the final answer is a relatively simple combination of the problem input data. In other cases the model output skips steps, or trails off and then produces the correct final answer. We took a conservative approach and marked these as false positives. More generally, understanding false positives more deeply is an interesting area for further study!
>
> > How many times each are the 20 examples in 5.2 modified to test memorization?
> Each question has four modifications, one for each of the modification types listed in Appendix J2.
>
> **Limitations**
> > Are you planning to release the training data?
>
> The arXiv data is publicly available and can be downloaded [here](https://arxiv.org/help/bulk_data). We are unfortunately unable to release the math webpage dataset as it was developed by filtering a proprietary internal dataset. We have however shared all the details needed to reproduce the processing of the dataset, in Appendix B.
>
> > details on the contract work used to clean the dataset
>
> We did not use a crowdsourcing service for this work, and instead hired a team of expert contractors at competitive wages. We are unable to share compensation information for these contractors.

---

> > ### Comment · Reviewer_8LQc · 2022-08-09
> > **Thank you**
> >
> > Thank you for the detailed response and answering all of my questions.

---

### Official Review · Reviewer_3teb · 2022-07-11

**Rating:** 9
**Confidence:** 5
**Soundness:** 4 excellent
**Presentation:** 4 excellent
**Contribution:** 3 good

**Summary:**

This work proposes a large language model *Minerva* to solve scientific and mathematical questions in natural language and generate step-by-step solutions using correct  Latex notation. This model is trained on a high-quality data collection composed of scientific and mathematical information. This model got impressive results across a wide variety of quantitative reasoning tasks, including Precalculus, Geometry, and many more.



**Questions:**

1. Any reason behind the incorrect answers for the solutions that were short?
2. Can it derive math expressions instead of numbers as the final answer like solving an indefinite integral?

**Limitations:**

There is currently no automated mechanism to check the correctness of the model's answers, and the authors also acknowledge it. This work has no major negative societal impact, but people may use it to cheat on some STEM tests.

**Strengths And Weaknesses:**

* This paper provides a well-documented analysis of results and the reason behind the usage of metrics.
* The work will be a founding stone in making AI for tutoring. This can have a significant impact on students with fewer resources.
* The authors also addressed the issues of False positives and Memorization with a detailed investigation.

---

> ### Author Response · Authors · 2022-08-02
> **Response to reviewer 3teb**
>
> Thank you for your encouraging review of our submission. We appreciate your feedback, and address your questions below:
>
>
> > Any reason behind the incorrect answers for the solutions that were short?
>
> For solutions that were evaluated by a human labeler as being too short, it is typically the case that a calculation step is missing, or that the model restates the premise of the question and then immediately jumps to producing the incorrect answer. As an example of this failure mode, one problem in MATH is "In the row of Pascal's triangle that starts with 1 and then 10, what is the next number?". The 8B model responds with "In the row of Pascal's triangle that starts with 1 and then 10, the next number is 100. Final Answer: The final answer is 100."
>
> > Can it derive math expressions instead of numbers as the final answer like solving an indefinite integral?
>
> Yes, the model does return mathematical expressions when prompted for them. For example, when prompted with the MATH few-shot prompt followed by the problem "Compute the antiderivative of $f(x) = x^3$" the model responds "Solution: The antiderivative of $f(x)=x^3$ is $\frac{x^4}{4}+C$. I hope it is correct."  Several questions in the OCWCourses are of this form, rather than numerical answers.

---

> > ### Comment · Reviewer_3teb · 2022-08-09
> > **Thanks for the detailed responses**
> >
> > The authors answered most of my queries and there is no change in my rating.

---

### Author Response · Authors · 2022-08-02
**Updated revision - new results & improved performance**

We thank all the reviewers for their detailed and thoughtful comments.

We want to first reiterate our main finding: focusing on scale and data quality can unlock new and surprising reasoning capabilities in large language models, despite the lack of any inductive bias specific to mathematics in the model architecture. Please see our response to reviewer iAwH regarding the originality and significance of our work, as well as a summary of other insights from our paper.

Our model achieves SOTA results on a variety of quantitative reasoning benchmarks, with “no outside access to libraries, calculators, or other tools in performing the task” (reviewer 9sZQ). We are delighted that the reviewers seem to agree that these results are significant, for example commenting that “the reasoning skill demonstrated by the model is amazing” (reviewer iAwH) and “The work will be a founding stone in making AI for tutoring” (reviewer 3teb).

We have updated the paper including several new results:
 - We trained and evaluated a 540B parameter model using the same procedure described in the paper. It achieves SOTA performance on “all” the benchmarks used in the paper. In particular, on the most challenging dataset, MATH, it reaches the 50% accuracy milestone! Note the previously published SOTA is 6.9%.
 - As suggested, we evaluated the model on the EQUATE reasoning dataset, and found that our model achieves SOTA performance on this benchmark.
 - We evaluated the 62B parameter model on MMLU-STEM using majority voting, obtaining SOTA results.
 - We evaluated the model on basic arithmetic with few-shot prompting, and found that the model obtains strong performance (for example, it achieves over 80% accuracy on 8-digit addition). We included an analysis of the model’s mistakes on basic arithmetic in the appendix.
 - We included all of the “false positive” examples we found in the appendix, so that readers can have a more complete picture of the model’s capabilities and limitations.
 - We have further updated the paper to address reviewers’ specific concerns.

---

### Meta-Review · Area_Chair_9A2F · 2022-08-24

**Recommendation:** Accept
**Confidence:** Certain

**Metareview:**

This is a very strong solid work that extracts equations from scientific papers and show strong performance in mathematical reasoning.

**Award:**

No

---

### Decision · Program_Chairs · 2022-09-14

Accept